# Research on Method of Farmland Obstacle Boundary Extraction in UAV Remote Sensing Images

**DOI:** 10.3390/s19204431

**Published:** 2019-10-12

**Authors:** Hui Fang, Hai Chen, Hao Jiang, Yu Wang, Yufei Liu, Fei Liu, Yong He

**Affiliations:** 1College of Biosystems Engineering and Food Science, Zhejiang University, Hangzhou 310058, China; hfang@zju.edu.cn (H.F.); 21713061@zju.edu.cn (H.C.); 11413017@zju.edu.cn (H.J.); 18768129100@139.com (Y.W.); yufeiliu@zju.edu.cn (Y.L.); fliu@zju.edu.cn (F.L.); 2Key Laboratory of Agricultural Internet of Things, Ministry of Agriculture Rural Affairs, Shaanxi 712100, China

**Keywords:** UAV remote sensing, coordinate registration, template matching, boundary extraction

## Abstract

Aimed at the problem of obstacle detection in farmland, the research proposed to adopt the method of farmland information acquisition based on unmanned aerial vehicle landmark image, and improved the method of extracting obstacle boundary based on standard correlation coefficient template matching and assessed the influence of different image resolutions on the precision of obstacle extraction. Analyzing the RGB image of farmland acquired by unmanned aerial vehicle remote sensing technology, this research got the following results. Firstly, we applied a method automatically registering coordinates, and the average deviations on the X and Y direction were 4.6 cm and 12.0 cm respectively, while the average deviations manually by ArcGIS were 4.6 cm and 5.7 cm. Secondly, with an improvement on the step of the traditional correlation coefficient template matching, we reduced the time of template matching from 12.2 s to 4.6 s. The average deviation between edge length of obstacles calculated by corner points extracted by the algorithm and that by actual measurement was 4.0 cm. Lastly, by compressing the original image on a different ratio, when the pixel reached 735 × 2174 (the image resolution reached 6 cm), the obstacle boundary was extracted based on correlation coefficient template matching, the average deviations of boundary points I of six obstacles on the X and Y were respectively 0.87 and 0.95 cm, and the whole process of detection took about 3.1 s. To sum up, it can be concluded that the algorithm of automatically registered coordinates and of automatically extracted obstacle boundary, which were designed in this research, can be applied to the establishment of a basic information collection system for navigation in future study. The best image pixel of obstacle boundary detection proposed after integrating the detection precision and detection time can be the theoretical basis for deciding the unmanned aerial vehicle remote sensing image resolution.

## 1. Introduction

The obstacle detection method in the road can be divided into ultrasonic technology, machine vision technology and laser radar technology according to the sensor [1]. The machine vision technology refers to the fact that the vision sensor is used to acquire the image of the vehicle on the way in place of human eyes, then it processes the image by means of methods, including color threshold value division, edge detection and stereoscopic matching and gets relevant information of the obstacle [2,3,4,5]. This method is cheap, easy to operate and has no influence on the surrounding environment. However, the following two problems would arise when this method is applied to the obstacle detection in farmland. Firstly, part of or the whole obstacle would be hidden in the crop when the farmland image is taken by the vehicle-mounted camera. Secondly, various obstacles exist in farmland, which frequently are farm tools, the impounding reservoir, the telegraph pole, people and livestock and so on. These things vary greatly in shape and size with different colors and it is difficult to accurately recognize them by threshold division or boundary detection.

Remote sensing (RS) refers to non-contact and remote detection technology [6]. The image acquisition by means of unmanned aerial vehicle RS technology over the farmland can avoid the obstacle from not being captured by camera. In recent years, it has rapidly developed, and the technology is featured with operation flexibility, low cost, high temporal-spatial resolution, strong environment adaptability, labor-saving, high efficiency and less pollution to environment. In the field of agriculture, the application of unmanned aerial vehicle not only avoids the damage made by large agricultural machinery to the crop, but also addresses the safety concern in the mechanized production, which is gaining increasing popularity from farmers and receiving scholars’ attention home and abroad [7]. In 2002, the unmanned aerial vehicle of NASA, Pathfinder-Plus was used to monitor weed eruption, exposed irrigation and abnormal fertilization and other situations [8]. Zarcotejada P.J. analyzed the moisture during orange planting through unmanned aerial vehicles, providing support for reasonable irrigation saving [9]. Zhang used the texture analysis method to establish rules to identify corn seed production through decision tree [10]. Yao acquired a multispectral image of wheat of different nitrogen level, density and variety through unmanned aerial vehicle remote sensing and found a method to analyze nitrogen of wheat and growing features through image analysis. Gong proposed output estimation module of oilseed rape based on fully constrained mixed pixel analysis method and unmanned aerial vehicle remote sensing [11]. Overall, unmanned aerial remote sensing technology has been widely applied in acquiring agricultural information, monitoring crop condition, and analyzing the effect of fertilization, but few in navigation. In this research, we aimed to analyze the feasibility of UAV remote sensing for automatic navigation of agricultural machinery.

As for remote sensing images acquired by unmanned aerial vehicle, Geographic Information System (GIS) is used to specifically analyze them with machine learning methods. It is usually used to achieve registering of coordinates of unmanned aerial vehicle remote sensing image and the process of obstacle boundary extraction. Meanwhile, to plan routes for navigation using the information extracted from the remote sensing image, it is necessary to design an algorithm to automatically realize the process mentioned above. Template matching is an effective pattern recognition technology, which can directly reflect similarity between the image and the pattern, and thereby find the target in the image and determine its coordinate position [12]. As it is accurate, strong in noise immunity and easy to realize, it has managed to be applied in target detection. Kherchaoui used human face features to establish the pattern and match, accomplishing human face detection [13]. Zhe proposed a manuscript number recognition method based on pattern matching and artificial neural network, whose recognition accuracy of number 0 to 9 reached 99.6% [14]. Cheng proposed a target location method of mixed pattern matching and threshold division, which utilized pattern matching method to realize coarse position of target detection and then to realize fine position by threshold division [15,16]. In recent years, the research points at home and abroad have always been how to increase calculation speed of pattern matching, but few scholars have extended the application of the method. The detection of static obstacle in the image acquired by unmanned aerial vehicle remote sensing is generally proceeded before the agricultural machinery. The method of pattern matching is exactly applicable when the precision is relatively strictly required while the requirement of real-time is relatively low.

This research used unmanned aerial vehicle remote sensing technology to firstly acquire an image of target farmland. Section 1 introduces the background and research that has been done. Section 2 explains the materials like remote sensing platform and photography equipment and methods including affine transformation algorithm and template matching. Section 3 shows the image processing results based on ArcGIS, results of coordinate registration based on algorithm, results of obstacle boundary extraction based on improved template matching method and finally the effect of image resolution on the extraction of obstacle boundary where the minimum image pixel which can be applied into obstacle boundary extraction was analyzed. Section 4 makes a conclusion that the coordinate automatic matching and obstacle boundary automatic extraction algorithm designed in this study can be used to build a basic information acquisition system for navigation in the future, which lays a foundation for the development of path planning and obstacle avoidance functions.

## 2. Materials and Methods

### 2.1. Remote Sensing Platform and Photography Equipment

The ultra-low altitude remote sensing platform used in this study is an eight-rotor UAV independently developed by Zhejiang University [17]. Its physical chart is shown in Figure 1a, and the specific performance is shown in Table 1.

Considering the weight and the pixel, we choose Sony A7RII full-range micro-single camera (as shown in Figure 1b) on the platform. The camera can take about 300 still pictures with its battery. The camera lens is Sony E-mount lens, which can focus automatically or manually.

### 2.2. Layout of Experimental Environment and Acquisition of Data

The aerial image of the test field was collected in the field in the west of the campus on the morning of December 13, 2018. The weather was clear and the wind speed was low on that day. The layout of experimental environment is shown in Figure 2, in which the size of aluminum block is 20 cm × 20 cm × 5 cm and the thickness and radius of navigation landmark’s umbrella shaped surface is 3 mm and 6 cm.

After the experimental environment was set, the longitude and latitude coordinates of the 0–26 center point of the landmark need to be measured by the C94 M8P module of RTK satellite positioning system. The measuring time at each point was 10 s. Five data values were obtained per second. Finally, the average values of 50 measured values at each point were summarized.

After the latitude and longitude coordinates were measured, the RGB image of the experimental plot was taken by UAV. The interval of UAV photography was set to 1 s and 104 aerial photographs were acquired in the whole process. It is necessary to use image mosaic technology to synthesize a complete image for the next analysis process. This research was completed by software Agisoft Photoscan 1.2.6. The RGB image obtained by stitching a series of original photographic images was stored in TIF format with a size of 448 M, and the resolution of the image was about 1 cm. It can be read directly by ArcGIS software for subsequent operation.

### 2.3. Affine Transformation Algorithm

Assuming that XOY is a Cartesian coordinate system, xo’y is a physical coordinate system, the intersection angle between two coordinate systems is α, the transverse distance between the coordinate origin o’ and O is $A_{0}$ and the longitudinal translation distance is $B_{0}$. The scale of the physical coordinate system (i.e., the scale of the pictures taken in this study) is $m_{x}$ and $m_{x}$. According to the principle of graphics, the coordinate transformation formula is as follows:(1)$$X=A_{0}+A_{1}x-A_{2}y$$
(2)$$Y=B_{0}+B_{1}x+B_{2}y$$

In formula, $A_{1}=m_{x}\cos\alpha$, $A_{2}=m_{x}\sin\alpha$, $B_{1}=m_{y}\sin\alpha$, $B_{2}=m_{y}\cos\alpha$.

Assuming that $Q_{x}$ and $Q_{y}$ present the coordinate difference between control points and the transformation value according to the formula individually.
(3)$$Q_{x}=X-\left(A_{0}+A_{1}x-A_{2}y\right)$$
(4)$$Q_{y}=Y-\left(B_{0}+B_{1}x+B_{2}y\right)$$

According to the principle of least square method, two sets of equations can be obtained by taking the sum and minimum of the squares of $Q_{x}$ and $Q_{y}$ as follows.
(5)$$\{\begin{array}{l} \sum X=A_{0}n+A_{1}\sum x-A_{2}\sum y\\ \sum xX=A_{0}\sum x+A_{1}\sum x^{2}-A_{2}\sum xy\\ \sum yX=A_{0}\sum y+A_{1}\sum xy-A_{2}\sum y^{2} \end{array}$$
(6)$$\{\begin{array}{l} \sum Y=B_{0}n+B_{1}\sum x+B_{2}\sum y\\ \sum xY=B_{0}\sum x+B_{1}\sum x^{2}+B_{2}\sum xy\\ \sum yY=B_{0}\sum y+B_{1}\sum xy+B_{2}\sum y^{2} \end{array}$$

The coefficients can be obtained by solving Equations (5) and (6), so the transformation of whole graph can be determined.

### 2.4. Template Matching

(1) Standard correlation coefficient matching

Template matching algorithm can be realized through the function “matchTemplate” in OpenCV. According to the different matching values, there are six commonly used methods: square difference matching, standard square difference matching, correlation matching, correlation coefficient matching and standard correlation coefficient matching. From simple (square difference matching) to complex (correlation coefficient matching), the more accurate matching results can be obtained, the longer calculation time it takes. In order to obtain higher detection accuracy (according to the official document in OpenCV), the standard correlation coefficient matching method was used in this research.

Using correlation coefficient to measure the similarity between two vectors. Assuming that the target template is a 5 × 5 image, it can be regarded as a 25-dimensional vector, and each dimension is the gray value of a pixel point. Comparing this vector with each sub-region in the image, the process of finding out the sub-region with the largest standard correlation coefficient is standard correlation coefficient matching, as shown in Equation (7):(7)$$\rho \left(c,r\right)=\frac{\sum_{i=1}^{m}\sum_{j=1}^{n}\left(g_{i,j}-\bar{g}\right)\left({g^{\prime }}_{i+r,j+c}-{\bar{g}}^{\prime }\right)}{\sqrt{\sum_{i=1}^{m}\sum_{j=1}^{n}{\left(g_{i,j}-\bar{g}\right)}^{2}\times \sum_{i=1}^{m}\sum_{j=1}^{n}{\left({g^{\prime }}_{i+r,j+c}-{\bar{g}}^{\prime }\right)}^{2}}}$$
where g(x,y) is the image gray function, (i,j) is the center pixel coordinates of the target window, $g^{\prime }\left(x,y\right)$ is the template gray function, and (i+r,j+c) is the center pixel coordinates of the search window. With $\bar{g}=\frac{1}{mn}\sum_{i=1}^{m}\sum_{j=1}^{n}g_{i,j}$, ${\bar{g}}^{\prime }=\frac{1}{mn}\sum_{i=1}^{m}\sum_{j=1}^{n}g_{i+r,j+c}$, we can get Equation (8):(8)$$\rho \left(c,r\right)=\frac{\sum_{i=1}^{m}\sum_{j=1}^{n}(g_{i,j}\times {g^{\prime }}_{i+r,j+c})-\frac{1}{mn}(\sum_{i=1}^{m}\sum_{j=1}^{n}g_{i,j})\left(\sum_{i=1}^{m}\sum_{j=1}^{n}{g^{\prime }}_{i+r,j+c}\right)}{\sqrt{\left[\sum_{i=1}^{m}\sum_{j=1}^{n}g_{i,j}{}^{2}-\frac{1}{mn}{\left(\sum_{i=1}^{m}\sum_{j=1}^{n}g_{i,j}\right)}^{2}\right]\left[\sum_{i=1}^{m}\sum_{j=1}^{n}g_{i+r,j+c}{}^{2}-\frac{1}{mn}{\left(\sum_{i=1}^{m}\sum_{j=1}^{n}g_{i+r,j+c}\right)}^{2}\right]}}.$$

With using template as the search window, the correlation coefficients of the original image are matched according to the fixed step size (usually 1 pixel). The closer the result is to 1, the higher the similarity between the region and the template is.

(2) Scale-Invariant Feature Transform (SIFT) descriptor matching

SIFT [18] features are invariant to rotation, scale scaling, brightness and so on. It is very stable for features extraction and mainly composed of the following four steps:

a. Extremum detection in Difference of Gauss (DOG) scale space: First, constructing DOG scale space, and using Gauss ambiguity of different parameters to express different scale spaces in SIFT. The construction scale space is used to detect the feature points that exist at different scales. 

b. Delete unstable extreme points. Two main types are deleted: low contrast extremum points and unstable edge response points.

c. Determine the main direction of feature points. The magnitude of the gradient of each pixel are calculated in the field with the feature point as the center and the radius of 3 × 1.5, and then the magnitude of the gradient is counted by histogram. The horizontal axis of the histogram is the direction of the gradient, and the vertical axis is the cumulative value of the gradient magnitude corresponding to the gradient direction. The direction corresponding to the highest peak in the histogram is the direction of the feature.

d. Generate descriptors of feature points. Firstly, the coordinate axis is rotated as the direction of the feature points, and the gradient magnitude and direction of the pixels in the 16 × 16 window centered on the feature points are divided into 16 blocks, each of which is the histogram statistics of eight directions in its pixels. A total of 128-dimensional feature vectors can be formed.

After getting the key points of the two images, we could match those feature points by calculating their distances. And the size and average distance of the top 10 key-points were used in the matching method.
(9)$$Socre=\frac{\sum_{i=0}^{10}distance}{10}-size_{match}$$

The score represents the matching degree between the obstacle image and the UAV image; we could find the best matched area on the UAV image which contains the object similar to the obstacle image.

## 3. Results and Discussion

### 3.1. Image Processing Results Based on ArcGIS

Firstly, coordinate registration was performed on the remote sensing image obtained. The essence of coordinate registration was to establish the transformational relation between user coordinates and physical coordinates. The registration process was implemented by the tool of Georeferencing in ArcGIS, and the control points needed to be manually selected.

In this study, a total of four control points (the centers of the landmarks) were set. When using ArcGIS for matching coordinates, the coordinate system of the control points that had been input would default to Cartesian coordinates. If the latitude and longitude coordinates were directly input, a large error might occur. Map projection should be conducted before the registration. Gauss–Krüger Projection is generally used in geographic information system in Hangzhou, China [19] because Hangzhou’s low latitude and Gauss–Krüger Projection’s high accuracy.

The latitude and longitude coordinate of the control points was shown in Table 2, where L and B represented the longitude and latitude of the control points, respectively. The X and Y after the coordinate transformation represented the abscissa and ordinate in the plane coordinate, respectively. Since the unit of the plane coordinate was meter, the three digits after the decimal point were taken. The serial numbers correspond to those in Figure 2.

Another eight marker points different from the four points of registration were selected to obtain the geographical coordinates (X,Y), and the actual coordinates (X’,Y’) was obtained by the Gauss–Krüger Projection, as shown in Table 3.

It can be analyzed from Table 4 that the maximum deviation in the X direction between the geographical coordinate converted from the eight mark points registration map and the geographical coordinate converted from the actual latitude and longitude was 9.7 cm, and the average deviation was 4.6 cm; the maximum deviation in the Y direction between the two was 13.7 cm, and the average deviation was 5.7 cm. There are two reasons for the deviations:

(1) There was a certain deviation of the latitude and longitude between actual coordinates and measured coordinates b, including the positioning deviation of the C94 M8P module itself, and the deviation between the mobile station position and the actual center point of the obstacle;

(2) During the registration process and the accuracy inspection process, the center of the marker was judged by the naked eye of the tester. Although the accuracy was high, a certain error still occurred.

In this research, the water-storage wells along the farmland were used as the obstacles (as shown in Figure 3), and the corner points of the obstacles were manually extracted on the ArcGIS-registered images. The plane coordinate (X,Y) was obtained from the registration with the unit of m, which is the information of the position of the obstacle boundary. This was shown in Table 4. 

### 3.2. Results of Obstacle Boundary Extraction Based on Improved Template Matching Method

Based on the traditional method of standard correlation coefficient template matching, this research improved the algorithm according to the specific experimental conditions. The original image size was 4408 × 13,047, so it would take a lot of time if the search was done using the step length of one pixel each time. Thus, using the rough matching method to determine the approximate position of the obstacle was considered. In this research, the original image was searched in steps of ten pixels, and six regions of interest containing obstacles were obtained. The image coordinates of each region in the original image coordinate system were recorded separately. Then the accurate matching method to search for six regions of interest in steps of one pixel was used to obtain the boundary of the six obstacles, and the image coordinates in the respective regions were recorded. Finally, the obstacle boundary in the original image was marked by coordinate transformation to obtain the specific information about the position. The time required for the improved standard correlation coefficient algorithm matching was reduced from 12.2 s to 4.6 s. The specific process is shown in Figure 4.

Besides standard correlation coefficient template matching, we tried a matching method based on SIFT descriptor for rough matching on Figure 4. The matching results are shown in Figure 5. Since the SIFT matching will match the whole image and there are many other objects in the origin image, we travelled through the origin image with a window, and we calculated the scores between this window and the obstacle image, then we found the window that got the highest score (the green one) on the Figure 5. Thus, we could get the approximate position of the obstacle on the origin image.

The matching method based on SIFT showed a more stable matching result and required less input, but this method consumes too much time (ten times or more) compared with standard correlation coefficient template matching. Since these two matching methods were used for a rough matching, the accuracy of this method would not be affected as long as the obstacles were found correctly. So, we chose standard correlation coefficient template matching method to complete the following analysis in this experiment.

The extraction results based on standard correlation coefficient template matching method were shown in Table 5.

In order to compare the accuracy of the two methods for the extraction of obstacle boundary, we selected one side of the obstacle (the straight-line distance from point I to point IV in Figure 3) as the research object, and compared the corner coordinates of the obstacles obtained by the two methods with the actual measured results respectively. The results were shown in Table 6.

It can be concluded from Table 6 that the maximum deviation of the side length of the obstacle based on ArcGIS and the actual measured length was 9.6 cm, the minimum was 0 cm, and the average deviation was 4.7 cm. The maximum deviation of the side length of the obstacle based on template matching method and the actual measured length was 6.3 cm, the minimum was 1.1 cm, and the average deviation was 4.0 cm. Thus, the difference of the average deviation between the two methods was 0.7 cm, which was quite small. Therefore, it can be concluded that the template matching method for UAV remote sensing image to extract the obstacle boundary is more accurate and can be used in the obstacle avoidance module of the automatic navigation system.

### 3.3. Results of Coordinate Registration Based on Algorithm

The image processing of coordinate registration was achieved by C++ and OpenCV function library in VS environment. The specific steps were as follows:

(1) Get region of obstacles (ROI). As explained in Section 3.2, we use a template matching method to get an ROI of obstacles.

(2) Obtain the image coordinate of the registration marker. The specific flow was shown in Figure 5, wherein the method of extracting the center was to calculate the center of gravity of the white portion in the binary image. The pixel of the extracted center point was the image coordinate of the registration point, and the geographical coordinate of the registration point was obtained by Gauss–Krüger’s positive calculation of the measured latitude and longitude coordinate. 

(3) Calculate the affine transformation coefficients with four configuration fiducial points. The specific formulas were shown in Equations (1)–(6).

(4) Calculate the transformation of all image coordinates to geographic coordinates X, Y using the obtained parameters.

After the completion of registration, the same eight mark points as in Table 4 were selected to verify the matching accuracy. The results are shown in Table 7. It can be seen from Table 7 that the maximum deviation in the X direction between the geographical coordinate converted from the eight mark points registration map and the geographical coordinate converted by the actual latitude and longitude was 9.2 cm, and the average deviation was 4.6 cm; the maximum deviation in the Y direction between the two was 24.3 cm, and the average deviation was 12.0 cm. Compared with the results in Table 7, the accuracy of the registration results in the X direction was similar (the average deviation was 4.6 cm), but the accuracy of the automatic registration in the Y direction was not as good. The reason might be that the smear in UAV imaging led to the inconsistency between the image of the marker and the actual shape, especially in the Y direction. This caused a large deviation in the center extraction by the algorithm, thus the algorithm needed further improvement. However, compared with the manual registration of ArcGIS, the automatic registration using the algorithm was more time-saving and labor-saving, and was more suitable for automatic route planning in automatic navigation.

### 3.4. Effect of Image Resolution on the Extraction of Obstacle Boundary

In this research, the resolution of the original image was 4408 × 13,047 pixels. With such resolution, the image processing speed was relatively slow. Even though the improved template matching algorithm was used, the time required to extract the boundary was still greater than 2 s, which could affect the real-time information acquisition. When the resolution of the image is reduced, it is obvious that the processing time of the image will be shortened. However, the accuracy of detecting the obstacle boundary will also be lowered. Thus, the influence of the image resolution on the results will be discussed in this section.

The original image was compressed in OpenCV, and the ratios were 1/2, 1/4, 1/6, 1/8, 1/10, 1/12, respectively. Then the correlation coefficient template matching was applied to the extraction of the obstacle boundary in the image with reduced pixels after compression. The observation showed that when the resolution was reduced to 1/12, although the naked eye could clearly identify the obstacles in the image, the error rate of the obstacles extracted by the algorithm reached 50%. Therefore, we only recorded the time taken to process the image with the first five ratios and obtained the pixel coordinates of the midpoint I in the boundary, before comparing them with the original image. The results were shown in Figure 6 and Table 8.

It can be analyzed from the Figure 7 and Table 8 that as the image pixels were reduced, the time required for image processing was greatly reduced. When the pixel was 1/10 of the original one, the time to extract the boundaries of the six obstacles was only 2.6 s. However, the accuracy of the extraction results also decreased with the decrease of the pixel. This research took the boundary of the obstacle extracted from the original image as the standard. When reduced by 1/2, the average deviations of the point I of the six obstacles in the X and Y directions were only 0.22 cm and 0.43 cm. When reduced by 1/10, the average deviations reached 1.73 cm and 2.60 cm. There were two main reasons. The first is that the reduction of image resolution led to the increase of deviation in the coordinate registration process, and the second is that the detection error of algorithm increased as the resolution decreased. 

In the automatic navigation of agricultural machinery, the accuracy deviation is generally less than 2 cm [20]. Considering the detection accuracy and detection time, the 735 × 2174 pixels (1/6 of the original image with the resolution of about 6 cm) of the remote sensing image used in this study can be used to detect obstacles, and the average deviations of point I of the six obstacles in the X and Y directions are 0.87 cm and 0.95 cm individually, and the detection time was about 3.1 s.

## 4. Conclusions

Based on image processing and template matching technology, an automatic coordinate registration and an obstacle boundary extraction algorithm were designed. The results were compared with those manually completed by ArcGIS software and the following conclusions could be drawn as follows.

(1) The RGB image of farmland in the west area of campus was acquired by using Sony A7RII camera on the eight-rotor UAV. The coordinate registration and obstacle boundary extraction process were completed by using ArcGIS. The average deviation between the geographic coordinates converted from eight landmark registration maps and the actual geographic coordinates converted from longitude to latitude was 4.6 cm in the X direction and 5.7 cm in the Y direction.

(2) The designed algorithm realized the automatic coordinates registration. The average deviation of the geographic coordinates converted from the eight landmark registration maps was 4.6 cm in the X direction and 12.0 cm in the Y direction. 

(3) According to the specific situation, the traditional correlation coefficient template matching method was improved, and the image processing time was greatly reduced. Based on this, an automatic extraction algorithm of obstacle boundary was designed. The average deviation between the extracted edge length of obstacles based on template matching method and the actual measured edge length of obstacle was 4.0 cm.

(4) The original image was compressed, and the ratio was 1/2, 1/4, 1/6, 1/8 and 1/10. The image with reduced pixels was matched by correlation coefficient template to extract obstacle boundary. Compared with the original image processing result, it was concluded that when the pixels reached 735 × 2174 (resolution reached 6 cm), the mean deviations of the boundary points I were 0.87 cm and 0.95 cm in X and Y directions, respectively. The whole detection process took about 3.1 s.

In conclusion, the coordinate automatic matching and obstacle boundary automatic extraction algorithm designed in this study can be used to build the basic information acquisition system for navigation in the future, which lays a foundation for the development of path planning and obstacle avoidance functions. The optimal image pixels for obstacle boundary detection proposed in this research after considering detection accuracy and detection time comprehensively provides a theoretical basis for the selection of UAV remote sensing image resolution.

## Figures and Tables

**Figure 1 sensors-19-04431-f001:**
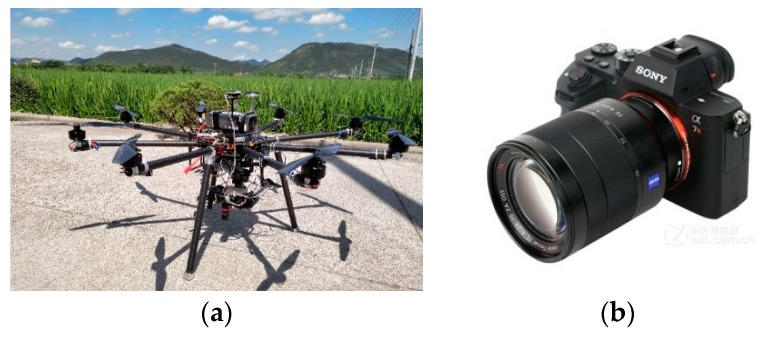
Sensing platform and photography equipment. (**a**) Eight-rotor UAV. (**b**) Sony A7RII full-range micro-single camera.

**Figure 2 sensors-19-04431-f002:**
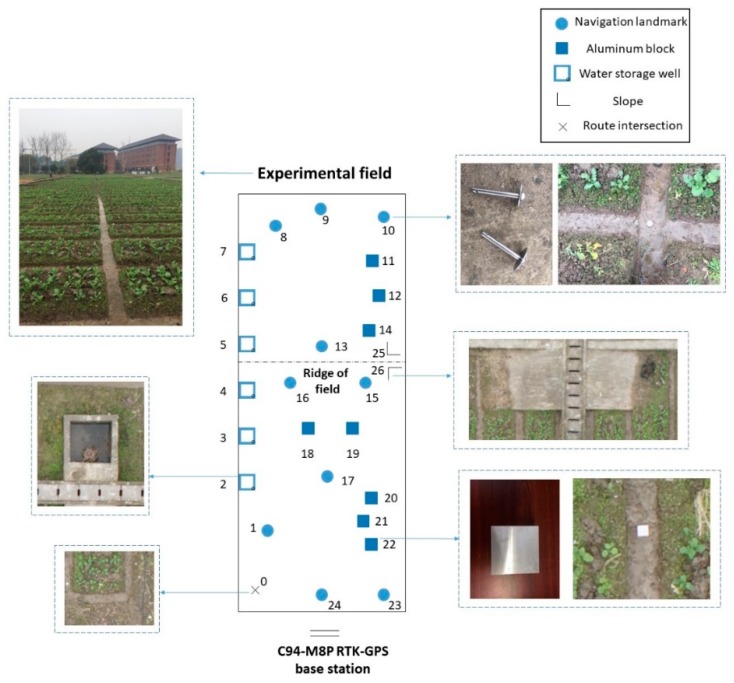
Diagram of experimental environment.

**Figure 3 sensors-19-04431-f003:**
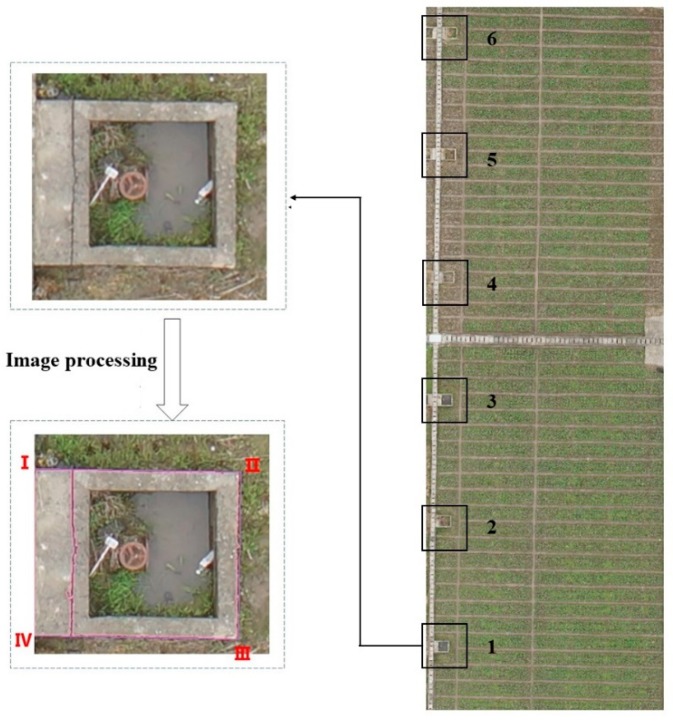
Obstacle number and corner sequence.

**Figure 4 sensors-19-04431-f004:**
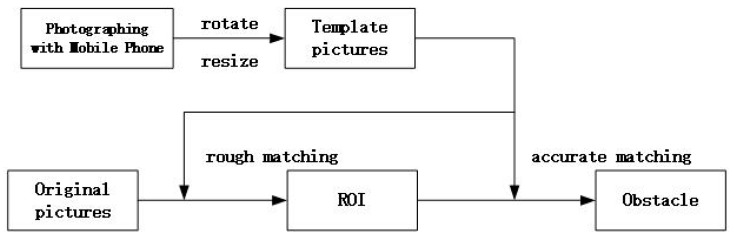
Flow chart of obstacle boundary extraction based on correlation coefficient template matching.

**Figure 5 sensors-19-04431-f005:**
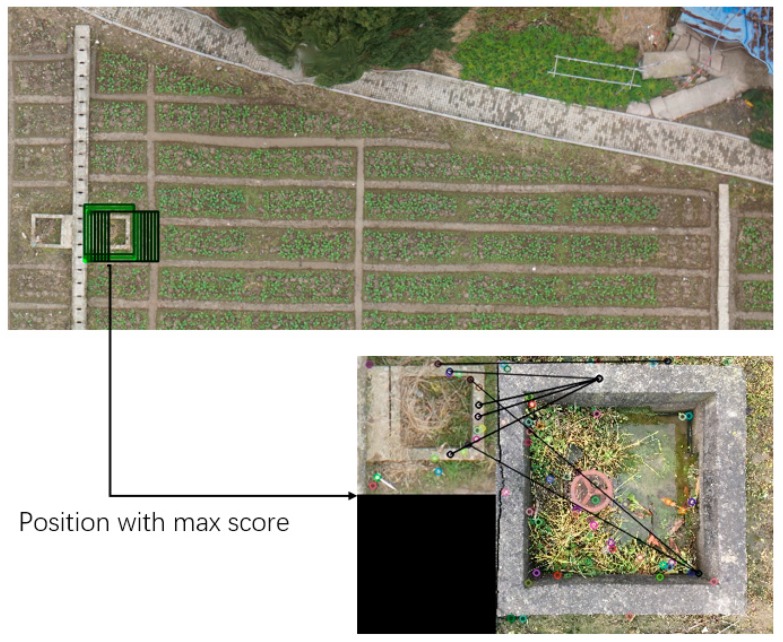
Travel through UAV image to find be best matched area.

**Figure 6 sensors-19-04431-f006:**
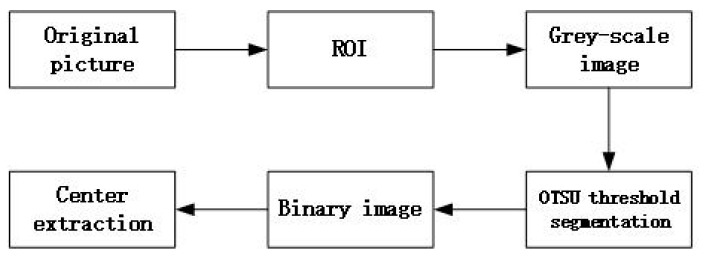
Flow chart of center extraction of registration markers.

**Figure 7 sensors-19-04431-f007:**
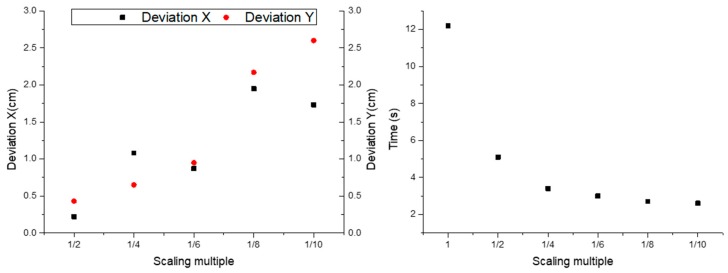
Boundary point I extraction from different pixel images.

**Table 1 sensors-19-04431-t001:** Parameters of eight-rotor unmanned aerial vehicle (UAV).

Performance	Parameter	Performance	Parameter
Fuselage diameter	1.1 m	Max-load	8 kg
Fuselage height	0.35 m	Max-altitude	500 m
Fuselage weight	3.5 kg	Max-endurance	25 min
Material	Carbon fiber	Remote sensing platform	Three-axis brushless cloud platform

**Table 2 sensors-19-04431-t002:** Latitude and longitude of control points and their plane coordinates.

Number	*L*	*B*	*X*	*Y*
0	30.3084806	120.0754564	7257.886	3,354,312.445
8	30.3083855	120.0746741	7182.646	3,354,301.845
10	30.3085322	120.0746454	7179.836	3,354,318.106
23	30.3086676	120.0754194	7254.313	3,354,333.169

**Table 3 sensors-19-04431-t003:** Accuracy analysis of ArcGIS registration results.

Number	Geographic Coordinates					
	*X*/m	$X^{\prime }/m$	deviation/m	*Y*/m	$Y^{\prime }/m$	deviation/m
9	7180.247	7180.296	0.049	3,354,306.324	3,354,306.461	0.137
11	7182.188	7182.167	0.021	3,354,313.827	3,354,313.954	0.127
12	7191.441	7191.398	0.043	3,354,317.348	3,354,317.397	0.049
13	7211.984	7211.928	0.056	3,354,311.750	3,354,311.775	0.025
17	7234.984	7234.991	0.007	3,354,316.475	3,354,316.483	0.008
19	7225.601	7225.668	0.067	3,354,320.290	3,354,320.272	0.018
21	7237.917	7237.820	0.097	3,354,328.555	3,354,328.599	0.044
22	7242.751	7242.724	0.027	3,354,330.839	3,354,330.795	0.044
Average			0.046			0.057

**Table 4 sensors-19-04431-t004:** Corner coordinates of obstacles extracted by ArcGIS.

Number	Geographic Coordinates							
	Ⅰ		Ⅱ		Ⅲ		Ⅳ	
	*X*/m	*Y*/m	*X*/m	*Y*/m	*X*/m	*Y*/m	*X*/m	*Y*/m
**1**	7247.542	3,354,307.763	7247.222	3,354,309.348	7248.532	3,354,309.566	7248.834	3,354,307.936
**2**	7234.359	3,354,305.454	7234.110	3,354,307.096	7235.415	3,354,307.292	7235.694	3,354,305.686
**3**	7221.311	3,354,303.263	7221.321	3,354,304.900	7222.707	3,354,305.038	7222.694	3,354,303.436
**4**	7208.761	3,354,301.172	7208.563	3,354,302.577	7209.853	3,354,302.787	7210.169	3,354,301.197
**5**	7195.928	3,354,298.716	7195.727	3,354,300.303	7197.023	3,354,300.535	7197.230	3,354,298.947
**6**	7183.296	3,354,296.327	7182.985	3,354,298.010	7184.296	3,354,298.153	7184.602	3,354,296.614

**Table 5 sensors-19-04431-t005:** Coordinates of obstacles extracted by template matching algorithm.

Number	Geographic Coordinates							
	Ⅰ		Ⅱ		Ⅲ		Ⅳ	
	*X*/m	*Y*/m	*X*/m	*Y*/m	*X*/m	*Y*/m	*X*/m	*Y*/m
**1**	7247.521	3,354,307.789	7247.228	3,354,309.373	7248.520	3,354,309.621	7248.812	3,354,308.037
**2**	7234.381	3,354,305.413	7234.074	3,354,307.077	7235.405	3,354,307.333	7235.712	3,354,305.669
**3**	7221.383	3,354,303.177	7221.383	3,354,304.801	7222.714	3,354,305.057	7222.677	3,354,303.433
**4**	7208.822	3,354,300.910	7208.523	3,354,302.520	7209.867	3,354,302.778	7210.166	3,354,301.168
**5**	7195.968	3,354,298.644	7195.677	3,354,300.214	7197.014	3,354,300.471	7197.305	3,354,298.901
**6**	7183.303	3,354,296.290	7182.999	3,354,297.927	7184.330	3,354,298.183	7184.633	3,354,296.546

**Table 6 sensors-19-04431-t006:** Barrier boundary length obtained by different extraction methods.

Method	Length/m						
	1	2	3	4	5	6	Average
ArcGIS extraction	1.304	1.355	1.393	1.408	1.322	1.337	1.353
Template matching extraction	1.315	1.355	1.319	1.369	1.361	1.354	1.346
Actual measurement	1.304	1.309	1.311	1.312	1.298	1.302	1.306
Deviation /ArcGIS	0	0.046	0.082	0.096	0.024	0.035	0.047
Deviation / template matching	0.011	0.046	0.008	0.057	0.063	0.052	0.040

**Table 7 sensors-19-04431-t007:** Analysis of automatic registration results.

Number	Geographic Coordinates					
	*X*/m	$X^{\prime }/m$	deviation/m	*Y*/m	$Y^{\prime }/m$	deviation/m
9	7180.254	7180.296	0.042	3,354,306.218	3,354,306.461	0.243
11	7182.185	7182.167	0.018	3,354,313.729	3,354,313.954	0.225
12	7191.409	7191.398	0.011	3,354,317.274	3,354,317.397	0.123
13	7211.979	7211.928	0.051	3,354,311.702	3,354,311.775	0.073
17	7234.908	7234.991	0.083	3,354,316.483	3,354,316.483	0
19	7225.714	7225.668	0.046	3,354,320.142	3,354,320.272	0.130
21	7237.912	7237.820	0.092	3,354,328.520	3,354,328.599	0.079
22	7242.751	7242.724	0.027	3,354,330.881	3,354,330.795	0.086
Average			0.046			0.120

**Table 8 sensors-19-04431-t008:** Boundary point I extraction from different pixel images.

Scaling Multiple	Average Deviation in X Direction/cm	Average Deviation in Y Direction/cm	Image Processing Time/s
1			12.2
1/2	0.22	0.43	5.1
1/4	1.08	0.65	3.4
1/6	0.87	0.95	3
1/8	1.95	2.17	2.7
1/10	1.73	2.60	2.6

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
