# Peer review of "Research on Method of Farmland Obstacle Boundary Extraction in UAV Remote Sensing Images"

_sensors, 2019, doi:10.3390/s19204431_

Round 1
Reviewer 1 Report
This paper illustrates a method based on the use of unmanned aerial vehicle for remote sensing, which enables to acquire image of target farmland and obstacle boundary extraction.
The authors have provided a revised version of their manuscript, and a better review of the literature was realized.
The subject of the paper and its results are suitable for the Journal.
For this revised version, the authors have clarified their results, and also included additional quantitative results, which are relevant for the understanding of its content. Also they have pointed out opportunities for real application, and explained well the reason to use image compression, and technical advantages. Likewise, they have provided additional explanations in terms of enhancements in the traditional method that uses the ArcGIS.
Nevertheless, for this new version the authors still should take into account some changes in Figure 6. In such context, the points presented on the three graphs cannot be connected to each other by lines, since they were obtained as a result of Deviation of X(cm) and Y(cm) versus Scaling Multiple, as well as Time(s) versus Scaling Multiple. Therefore, only the points must remain on the graphics, i.e., without the use of lines.
Author Response
Thank you for your advices. I’ve removed the lines on Figure 6.
Reviewer 2 Report
All comments have been addressed.
Author Response
Thank you.
Reviewer 3 Report
My comments and questions are given in the attached file.

Author Response
Thanks for your advices and here are my responses:
We tried an image matching method base on sift descriptor. And it showed a stainable result than the simple template matching method although it required much more time than the template matching method. We considered of applying machine learning on our template matching method but it will require a lot of samples and it might lack of real-time ability, so we dropped that idea for now and maybe pay more attention on it on our next experiment. With the image matching method base on sift descriptor, we don’t need to resize the template image to the same size as the that in the origin image which is more automatic. I understand your concern, but we thought our main contribution is not the improvement on the algorithms for the image processing techniques but the idea and identification, in some degree, of using those technics for obstacles detecting with UAV images, some references for deciding the image resolution on UAV and remote sensing and registering coordinates.The sift descriptor matching method will be added on the latest resubmission.
thank you.
Hai CHEN
Round 2
Reviewer 3 Report
My concerns have been clarified in the revised version of the paper and the author's reply. The SIFT-based matching method, added to the manuscript, seems to be invariant on the obstacle size. I believe the manuscript has been improved.
This manuscript is a resubmission of an earlier submission. The following is a list of the peer review reports and author responses from that submission.
Round 1
Reviewer 1 Report
Manuscript: Research on Method of Farmland Obstacle Boundary Extraction in UAV Remote Sensing Images
This paper illustrates a way to construct a method based on the use of unmanned aerial vehicle for remote sensing to acquire image of target farmland. The authors have shown the opportunity to have an algorithm considering an automatically registered coordinate and obstacle boundary extraction. Besides they claim that results were compared with other based on the use of ArcGIS. Finally, they have used image compression and a template matching algorithm to figure out the minimum image pixel for obstacle boundary extraction and analysis. However, the author’s didn’t prepare an adequate review of the literature.
The subject of the paper is interesting and suitable for the Journal.
The results need to be better clarified, and some additional quantitative results should be pointed out by the authors. Therefore, the way in how to use such possible method should be stablished, also when considering the possibility of such use for real application. Besides, the authors should explain the criteria used for the camera selection, and the bases to select the number of images.
The authors should explain where they have found enhancements in the traditional method based on the use of the ArcGIS, and if they have observed any comparison with other published method, or even data, in the literature used for agricultural obstacle boundary extraction.
The authors should explain if there is any possibility (or not) to get the generalization of the method i.e., taking into account a calibration curve for typical agricultural obstacles, since their method requires a minimum of validation, i.e., which is mainly based on the pixel size.
The abstract should be rewritten, since the paper is not just a proposal but in fact the presentation of a process to carry out a method for farmland obstacle boundary extraction in UAV images
The template matching algorithm should be better explained. The author should observe when will be necessary to take in account constrains for the validation and use of equations (7) and (8).
In the section Results and Discussions a better observation about the use of references are required. In Figure 5 (Flow chart of obstacle boundary extraction based on correlation coefficient template matching) the authors should better explain how the ROI is selected, and how is its dependence in relation of the pixel size. The period “In the automatic navigation of agricultural machinery, the accuracy deviation is generally less than 2 cm [20]” could be used in the introduction.
In the Conclusion section the authors should observe in their paper why they have compressed the images considering ratios of 1/2, 1/4, 1/6, 1/8 and 1/10. How this contributed to the results? Furthermore, and finally, the authors should make it clear why the optimal image pixels for obstacle boundary detection can provide a theoretical basis for the selection of UAV remote sensing image resolution.
Reviewer 2 Report
The first paragraph of Section 1 is badly written. Please rewrite it. Thinking about the topic of your paper, the drawback of current method and the novelty of your research.
Please summarize the structure of the manuscript at the end of Section 1.
Line 100, what does the mean of “300 still pictures”?
Line 151, in order to obtain higher detection accuracy, the standard correlation coefficient matching method was used in this research. Why did you choose “standard correlation coefficient match”? Any experiments prove it has a higher detection accuracy?
Line 175, Gauss-Krüger Projection is generally used in geographic information system in China. Which projection system is chosen is decided by the requirement of the project, the area of study site, etc. Please demonstrate the reason why choose Gauss-Krüger Projection.
You used an automatic registration method for registration. I think some similar methods have been used in some other studies. Please summarize other studies which also used this algorithm into Section 1. Also demonstrate the novelty of your method compared with the method in those studies.
Line 233. You mentioned the reason to cause the large deviation in Y direction and the algorithm needs further improvement. How can you improve the algorithm?
Line 245. Why 10 pixels were selected as the step interval? Any experiment proves it is better than any other invervals?
Line 263, Table 8 should be Table 7, right?
Figure 6 (a), one legend should be Deviation Y, right?
Line 337. Wrote the draft of manuscript, right?
Many sentences are hardly to be understood. Proofreading by a native speaker is highly recommended.
Reviewer 3 Report
My comments and questions are given in the attached file.
